# Graph Knowledge Distillation to Mixture of Experts

**Pavel Rumiantsev**                                                    *pavel.rumiantsev@mail.mcgill.ca*
*The Department of Electrical and Computer Engineering*
*McGill University*

**Mark Coates**                                                         *mark.coates@mcgill.ca*
*The Department of Electrical and Computer Engineering*
*McGill University*

**Reviewed on OpenReview:** *https://openreview.net/forum?id=vzZ3pbNRvh*

## Abstract

In terms of accuracy, Graph Neural Networks (GNNs) are the best architectural choice for the node classification task. Their drawback in real-world deployment is the latency that emerges from the neighbourhood processing operation. One solution to the latency issue is to perform knowledge distillation from a trained GNN to a Multi-Layer Perceptron (MLP), where the MLP processes only the features of the node being classified (and possibly some pre-computed structural information). However, the performance of such MLPs in both transductive and inductive settings remains inconsistent for existing knowledge distillation techniques. We propose to address the performance concerns by using a specially-designed student model instead of an MLP. Our model, named Routing-by-Memory (RbM), is a form of Mixture-of-Experts (MoE), with a design that enforces expert specialization. By encouraging each expert to specialize on a certain region on the hidden representation space, we demonstrate experimentally that it is possible to derive considerably more consistent performance across multiple datasets. Code available at `https://github.com/Rufaim/routing-by-memory`.

## 1 Introduction

Graphs can be used to encode the dependencies between data samples. The impressive performance of Graph Neural Networks (GNNs) shows that taking into account the structural information increases the quality of prediction on tasks like product prediction on co-purchasing graphs or paper category prediction on citation graphs (Kipf & Welling, 2016; Hamilton et al., 2017). However, despite the potential accuracy improvements of GNNs, multi-layer perceptrons (MLPs) remain preferable to graph neural networks for many large-scale industrial applications. This is due to the fundamental inefficiency of GNNs, with scalability limitations making deployment challenging (Zhang et al., 2020a; Jia et al., 2020; Zheng et al., 2022). GNNs operate in layers, and each layer requires the processing of a neighbourhood of the node in order to compute the prediction. For example, evaluating the prediction for a single node with an $L$-layer GNN requires processing at least every node in the $L$-hop neighbourhood. For real-world graphs, involving millions of nodes, the $L$-hop neighbourhood can be very large, leading to resource intensive operations (Jin et al., 2021). Even if we only sample a subset of the neighbours at each layer, the receptive field for the node can grow very rapidly. This can lead to a latency performance bottleneck in high-load systems. This is especially the case in situations where the graph is distributed over multiple servers. Fetching the entire $L$-hop neighbourhood can result in greatly increased latency (Zheng et al., 2022). By contrast, forming a prediction for a single node with an $L$-layer MLP requires processing only the features of that node. This naturally eliminates the latency problems associated with additional node fetching. Therefore it can be scaled and deployed efficiently, and parallelization is straightforward.

Combining GNN's outperformance with the reduced latency of MLP allows us to enjoy the advantages of both (Zhang et al., 2021b). Early works tried to simplify the aggregation step of the GNN by decreasing the number of operations performed at inference (Hu et al., 2021; Zheng et al., 2021). However, such methods still depend on node fetching, which may seriously increase latency for large graphs (Jin et al., 2021; Zheng et al., 2022). Knowledge distillation is a more efficient way to address this problem. A student MLP is trained directly using soft labels generated by a teacher GNN, and can thus approximate the graph-context information obtained by the aggregation step. This leads to reduced latency and can even result in higher inference quality in some cases (Zhang et al., 2021b; Tian et al., 2022; Wu et al., 2023).

Increasing the number of parameters of the student MLP can help to achieve better performance for some datasets (Zhang et al., 2021b; Tian et al., 2022). However, this is not a consistent effect. In this work, we aim to improve the student performance for a fixed number of parameters by introducing a Routing-by-Memory (RbM) architecture as the student model. Our proposed model is a Sparse Mixture of Experts (MoE) (Shazeer et al., 2016; Chi et al., 2022) approach that is tailored to the graph learning setting. By avoiding the aggregation step and incorporating a sparse model structure, we can achieve higher parameter capacity, leading to better performance while keeping the inference cost low. The Routing by Memory (RbM) procedure encourages experts to specialize on a specific subset of the representations, making it more efficient than standard MoE routing. We conduct a series of experiments showing that our approach can be efficiently and effectively applied to datasets of various sizes. To evaluate our model, we explore both transductive and inductive settings for 9 publicly available datasets.

We make the following contributions:

- Ours is the first work to propose the use of a student Mixture-of-Experts (MoE) model for the distillation of a GNN.

- We propose a Routing-by-Memory (RbM) model that differs from, and outperforms, a standard sparse MoE. We introduce important adaptations to a routing system previously proposed by Zhang et al. (2021a) for routing to a single expert in a computer vision setting. We allow for routing to multiple experts and use a different distance.

- During training of the proposed MoE, we introduce several loss terms to encourage better clustering of representations and improved expert specialization.

- We conduct multiple novel experiments, demonstrating that the proposed approach consistently outperforms both (i) enlarged MLP students; and (ii) ensembles or sparse MoEs.

## 2 Related work

### 2.1 GNN-to-MLP Knowledge Distillation

Knowledge distillation from a Graph Neural Network (GNN) into a Multi-Layer Perceptron (MLP) promotes inference efficiency by avoiding the aggregation over neighbourhood nodes. Yang et al. (2021) presented one of the first distillation attempts, employing a student model that combines label propagation with an MLP acting on the features. Although label propagation is a lightweight form of aggregation, it is still reliant on the graph, so the overall speed-up in inference time is not dramatic.

The GLNN by Zhang et al. (2021b) introduces knowledge distillation to an MLP without any aggregation over the graph nodes. The student is trained with node content features as input and soft labels, generated by a pretrained GNN, which acts as the teacher. Tian et al. (2022) show that soft labels alone are not enough to achieve consistent performance due to noise injected by the teacher GNN. The presented NOSMOG approach incorporates a set of techniques to be used on top of knowledge distillation to help the student MLP to better approximate the graph-based information. It includes explicitly encoded position features generated using DeepWalk (Perozzi et al., 2014). NOSMOG also introduces a representational similarity distillation, which strives to encourage the student MLP to preserve the similarities between node representations that are observed in the teacher GNN. Adversarial attack perturbations are also applied in order to ensure that the student model is more stable. While all the introduced techniques provide some improvement, the positional encoding is responsible for the vast majority of the performance gain.

Zhang et al. (2020b) present an approach that aims to estimate the quality, or reliability, of the teacher model soft labels by evaluating the entropy. Tan et al. (2022) also aim to evaluate the reliability of the soft labels, but employ a reinforcement learning approach. The concept of reliability is also at the heart of the work by Wu et al. (2023), who propose KRD, a method that regularizes the student MLP by making it predict the soft labels of a subset of the neighbouring nodes as well as that of the local target node. The subset of neighbours is selected according to how reliable their associated soft labels are estimated to be. This regularization strategy is very effective, especially when combined with explicit positional encodings.

## 2.2 Mixture-of-Experts

A Sparse Mixture-of-Experts (MoE) model is a weighted combination of similarly structured models with dynamically computed weights (Shazeer et al., 2016; Gross et al., 2017; Zhang et al., 2021a; Li et al., 2022; Dryden & Hoefler, 2022; Chi et al., 2022; Komatsuzaki et al., 2022; Pavlitska et al., 2023). For any sample, only a small portion of the experts have non-zero weights. This allows us to increase the model learning capacity without significantly inflating the processing costs (Fedus et al., 2022). An MoE can also be used as a layer inside a larger model (Qu et al., 2022; Yan & Li, 2023). In most implementations, the computation of the expert weights, referred to as routing, is performed by a separate neural network (a policy network) of a size comparable to an expert. Based on the MoE input, it produces weights for the experts. Because both the policy network and expert networks are trained simultaneously, there is a danger of under-utilization, and consequently under-training, of some experts (Krishnamurthy et al., 2023). This is commonly mitigated by injecting noise and designing loss functions that even out the utilization.

An alternative routing scheme, designed to prevent routing inconsistencies, involves pairing each expert network with an embedding vector. Instead of predicting the expert weights directly, the policy network then aims to project the input sample into the embedding space. The weights are derived from the distances to the expert embeddings (Gross et al., 2017; Zhang et al., 2021a; Chi et al., 2022; Li et al., 2022; Yan & Li, 2023; Qu et al., 2022). Initially, the Euclidean distance was used to route to the single closest expert (Gross et al., 2017; Zhang et al., 2021a). However, this causes computational stability issues when multiple experts are used (Qu et al., 2022). Using dot-product similarity (Lample et al., 2019; Fedus et al., 2022) instead of Euclidean distance typically leads to representation collapse. Chi et al. (2022) used cosine similarity to avoid the collapse, and this is currently used in many implementations (Li et al., 2022; Yan & Li, 2023). Zhang et al. (2021a) propose moving the expert embeddings into the layer's input space in order to encourage expert specialization. In their scheme, the input vector is always routed to a single expert, the one with the closest embedding vector. Each expert embedding is updated by calculating a moving average over the input embedding vectors that are routed to that expert. This leads to each expert specialising on the area of the input space around its embedding.

Knowledge distillation into a Mixture-of-Experts has not been intensively studied. In tangentially related work, Zuo et al. (2022) study the distillation of language models and incorporate the MoE structure into pre-trained models for fine-tuning. Komatsuzaki et al. (2022) explore the task of upgrading a pre-trained dense model into a larger, sparsely-activated MoE.

## 3 Background

We denote a graph by $\mathcal{G} = (\mathcal{V}, \mathcal{E}, \mathbf{X})$, where $\mathcal{V}$ is a set of $N$ nodes, $\mathcal{E}$ is a set of edges between nodes, and $\mathbf{X} \in \mathbb{R}^{N \times d}$ represents a matrix with each row being a vector of $d$ node features associated with the corresponding node. For the node classification task, with $C$ classes, we use a label matrix, $\mathbf{Y} \in \{0, 1\}^{N \times C}$, with each row containing a one-hot encoded class label. The superscript $L$ denotes the labelled nodes of the graph and the superscript $U$ denotes unlabelled nodes, i.e., $\mathcal{V}^L$, $\mathbf{X}^L$, $\mathbf{Y}^L$ are, respectively, the labeled nodes, their node features, and the one-hot class labels.

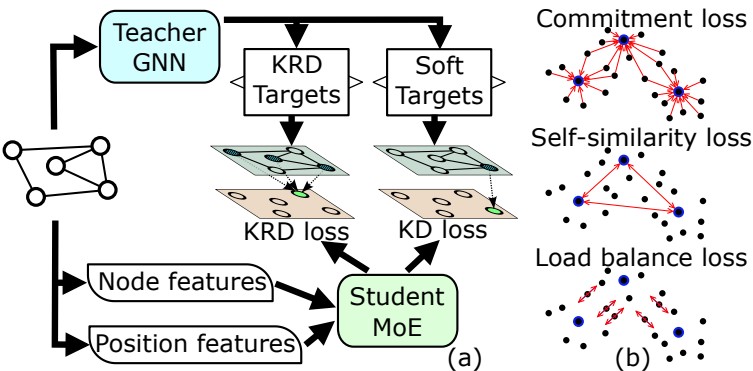

Figure 1: (a) An overview of the overall training framework. A teacher GNN is trained on the graph and provides targets for Knowledge Distillation (KD) (9) and Knowledge-Aware Reliable Distillation (KRD) (12) losses. A Mixture-of-Experts student is trained on the node features and positional encoding (see Section 4.2). (b) We use three additional losses to adjust the internal representations of the model, as the embeddings we use for routing (see Section 4.1). We provide schematic representations of these losses to aid intuitive understanding. Commitment loss (6) pulls representations closer to embeddings (highlighted in blue). Self-similarity loss (7) prevents collapse of representations. Load balance loss (8) helps to move borderline representations towards embedding of the less populated experts.

## 4 Methodology

We now introduce our distillation approach, which uses a Mixture-of-Experts (MoE) model. The method starts with the training of a teacher GNN. The teacher model is used to produce soft-labels for the knowledge distillation (see Section 4.2). The knowledge distillation setup uses a combination of reliable sampling and positional encoding. Our student model is a Routing-by-Memory model, with a special routing procedure that enforces expert specialisation (see Section 4.1). Section 4.3 describes the expert initialization procedure.

Figure 1 provides an illustration of the overall training framework. A teacher GNN is trained on the graph and provides soft targets for Knowledge Distillation (KD) (9) and Knowledge-Aware Reliable Distillation (KRD) (12) losses. The student model is trained with the node features and positional encodings as inputs. We introduce spatial routing by memory, so at each layer, each expert is represented by an embedding in the same space as the input representations for that layer. Representations are then routed to the closest experts. Three types of loss terms, discussed in more detail below, are used to encourage effective learning of the expert embeddings and the hidden representations. Figure 1(b) depicts the goals of these losses. A commitment loss pulls representations closer to embeddings, encouraging specialization of experts. A self-similarity loss pushes the expert embeddings apart and prevents the collapse of the representations. A load balance loss strives to achieve more equal utilization of experts by moving representations that are almost equidistant to two or more experts closer to the less-utilized experts.

### 4.1 Spatial routing by memory

We use the standard formulation for a Mixture-of-Experts layer, introduced by Shazeer et al. (2016):

$$\mathbf{h}_l = \sum_{i=1}^{E} G(\mathbf{h}_{l-1})_i f_i(\mathbf{h}_{l-1}), \tag{1}$$

where $G : \mathbb{R}^{d'} \to [0,1]^E$ is a policy network that produces routing coefficients, $f_i : \mathbb{R}^{d'} \to \mathbb{R}^{d''}$ is an expert, $E$ is the number of experts, and $\mathbf{h}_{l-1} \in \mathbb{R}^{d'}$ is the input hidden representation of the datapoint, $\mathbf{x} \in \mathbb{R}^d$, emerging from the $(l-1)$-th layer. For the first layer, the hidden representation is the input, i.e., $\mathbf{h}_0 = \mathbf{x}$.

Li et al. (2022) introduce a routing scheme that uses a set of embeddings, $\mathbf{Q}^{MoE} \in \mathbb{R}^{E \times d_e}$, with each embedding being associated with a particular expert. The weight assigned to each expert for a given input

at layer $l$ is determined by measuring the cosine distance between the projection of the input vector $h_{l-1}$ and the expert's embedding. The policy network routes the input vector to the $k$ nearest experts:

$$G_{MoE}(\mathbf{h}) = \text{softmax}\left(\text{Top}_k\left(\frac{\mathbf{Q}^{MoE}\mathbf{Wh}}{\|\mathbf{Q}^{MoE}\|\|\mathbf{Wh}\|}\right)\right). \tag{2}$$

Here $\mathbf{W} \in \mathbb{R}^{d_e \times d'}$ is a trainable projection matrix, and the operation $\text{Top}_k(\cdot)$ is a one-hot embedding that sets all elements in the output vector to zero except for the elements with the largest $k$ values. Chi et al. (2022) demonstrate that using a policy network of this form results in a more even distribution of token projections over the projection space, leading to improved performance.

While Li et al. (2022) provide analysis demonstrating that $G_{MoE}(\cdot)$ provides a degree of specialization for expert networks, we found it insufficient (see the experimental results in Section 5.5). We therefore enforce experts' local specialization by setting the expert embeddings $\mathbf{Q}^{RbM} \in \mathbb{R}^{E \times d'}$ to vectors in the input space, rather than projecting to a separate space. We achieve this by setting:

$$G_{RbM}(h) = \text{softmax}\left(\text{Top}_k\left(\frac{\text{sg}[\mathbf{Q}^{RbM}]\mathbf{h}}{\|\text{sg}[\mathbf{Q}^{RbM}]\|\|\mathbf{h}\|}\right)\right), \tag{3}$$

where $\text{sg}[\cdot]$ is a stop gradient function. In our approach, each expert embedding vector is positioned at the center of an area in the representation space that the expert is specialising on. We use cosine similarity and interpolation between multiple experts (see Figure 2).

As every embedding vector in our approach can be interpreted as a centre of a cluster, we do not update it with gradient descent, but instead use a direct approach, calculating a moving average over the input batches. We evaluate:

$$\mathbf{Q}_i^{RbM}(t+1) = \hat{\lambda}(t)\mathbf{Q}_i^{RbM}(t) + (1-\hat{\lambda}(t))\sum_{\substack{j=1 \\ G_{RbM}(\mathbf{h}_j)_i \neq 0}}^{B} \text{softmax}_j(G_{RbM}(\mathbf{h}_j)_i)\,\mathbf{h}_j\,, \tag{4}$$

where $\hat{\lambda}(t) = \max(\lambda(t), 1.0)$ is a coefficient. We use an annealing schedule to ensure that $\lambda(t) < \lambda(t+1)$. As $t \to \infty$, the change in the expert embeddings tends to zero, facilitating the convergence of all embeddings.

We compute this sum only for the vectors that are routed to the expert. Applying the softmax over the batch dimension improves performance.

In order to enhance training we use expert-wise constant attention for input and output. Thus the entire Mixture of Experts block can be formulated as:

$$\mathbf{h}' = \exp(s)\sum_{i=1}^{E} G_{RbM}(\mathbf{h})_i f_i(\exp(\mathbf{att}_i) \odot \mathbf{h}), \tag{5}$$

where $s \in \mathbb{R}$ is a learnable output scaler, $\mathbf{att}_i \in \mathbb{R}^d$ is a learnable input attention vector, and $\odot$ denotes element-wise multiplication. Note that the attention is applied only to the expert input and not to the router input. This routing approach is inspired by the technique presented by Zhang et al. (2021a); we extend it to perform top-k rather than top-1 routing and employ cosine similarity instead of Euclidean distance.

In order to encourage tighter clustering of hidden representations around the expert embeddings we incorporate a vector quantization (VQ)-style commitment loss:

$$Loss_{VQ} = -\frac{1}{B}\sum_{i=1}^{B}\sum_{j=1}^{E} G_{RbM}(\mathbf{h}_i)_j \frac{\text{sg}[\mathbf{Q}_j^{RbM}]\mathbf{h}_i}{\|\text{sg}[\mathbf{Q}_j^{RbM}]\|\|\mathbf{h}_i\|}\,, \tag{6}$$

where $B$ is the batch size. When MoE layers are stacked, this loss encourages the hidden representations to move closer to the nearest expert embeddings. It prevents frequent fluctuations in routing and allows experts

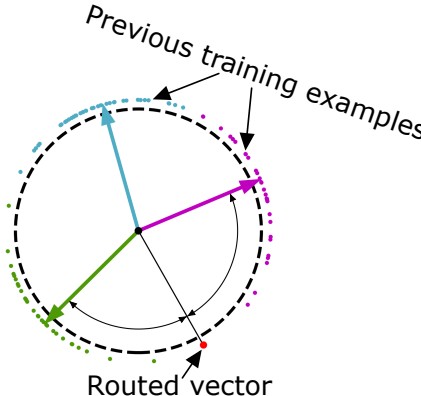

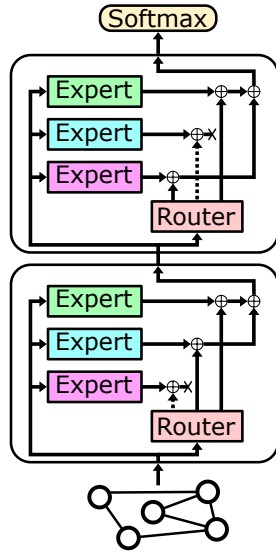

Figure 2: A simplified example of cosine routing (3). Three experts are present in total ($E = 3$). Two experts are used at a time ($k = 2$), and thus the two experts with closest embeddings are used. Arrows show expert embeddings on the unit circle. Points are representations of the previously routed training examples (see equation 4).

Figure 3: Schematic depiction of a student model with two RbM layers. Three experts are present per layer with two experts used for each sample.

to acquire specialization. Routing to multiple experts prevents the hidden representations from collapse. A similar loss was presented by Razavi et al. (2019) for the image generation task in computer vision, and here we adapt it for cosine similarity.

We incorporate a self-similarity loss in order to additionally spread embeddings on the hypersphere:

$$Loss_{SS} = \frac{1}{E^2} \sum_{i=1}^{E} \sum_{j=1}^{E} \frac{\text{sg}[\mathbf{Q}_j^{RbM}]\mathbf{Q}_i^{RbM}}{\|\text{sg}[\mathbf{Q}_j^{RbM}]\|\|\mathbf{Q}_i^{RbM}\|} . \tag{7}$$

For this loss, we allow a gradient descent update of the expert embeddings. Gradients are propagated only through one part of the computation, by using the stop-gradient, to prevent instability.

Finally, we apply a load balance loss, as proposed by Shazeer et al. (2016):

$$Loss_{LB} = \frac{Var(\mathbf{Load})}{Mean(\mathbf{Load})^2}, \qquad \text{for} \quad \mathbf{Load} = \sum_{i=1}^{B} G_{RbM}(\mathbf{h}_i). \tag{8}$$

Here $\mathbf{Load} \in \mathbb{R}^E$ is the vector of unnormalized expert utilization. While expert specialization does not require load balancing, we adopt this loss to redistribute input vectors that lie in almost equal proximity to multiple experts. This loss encourages representations to move closer to the least utilised expert embeddings.

## 4.2 Knowledge Distillation

We distill knowledge from the pretrained GNN using supervised learning, regularized by the KL-divergence between the class distribution $\hat{y}_v$ predicted by the student and the class distribution $\hat{y}'_v$ predicted by the teacher. The knowledge distillation loss combines cross-entropy (CE) with KL-regularization:

$$Loss_{KD} = \frac{\nu}{|\mathcal{V}^L|} \sum_{v \in \mathcal{V}^L} CE(\hat{\mathbf{y}}_\mathbf{v}, \mathbf{y}_\mathbf{v}) + \frac{1-\nu}{|\mathcal{V}|} \sum_{\mathbf{v} \in \mathcal{V}} \mathbf{KL}(\hat{\mathbf{y}}_\mathbf{v}, \hat{\mathbf{y}}'_\mathbf{v}) . \tag{9}$$

Here $\nu$ is a hyperparameter that controls how much the knowledge distillation training relies on the soft labels from the teacher GNN, $CE(\cdot, \cdot)$ denotes cross-entropy, and $KL(\cdot, \cdot)$ is a KL-divergence (Zhang et al.,

2021b). While the supervised part of the loss uses only the labeled part of the graph, the KL-divergence is computed over the entire set of nodes.

We use positional encodings generated by DeepWalk (Perozzi et al., 2014) in our distillation procedure. Before running the student training we learn positional encodings by running the DeepWalk algorithm on the input graph. The positional encodings are based solely on the graph structure, and are concatenated with the node features when nodes are processed by the student model.

In order to additionally leverage the graph structure, we employ the knowledge-based sampling technique of KRD (Wu et al., 2023). KRD is based on the principle of aligning the representation of a target node with those of *reliable* nodes in its neighbourhood. To accomplish this, Wu et al. (2023) introduce a measure of reliability, which is based on identifying nodes whose teacher GNN representations do not change substantially when the features in the graph are subjected to perturbation. The reliability metric for node $j$ is:

$$\rho_j = \frac{1}{\delta^2} \, \mathbb{E}_{\tilde{\mathbf{X}} \sim N(\mathbf{X}, \delta\mathbf{I})} \left\| \mathcal{H}(\hat{\mathbf{y}}_j') - \mathcal{H}(\tilde{\mathbf{y}}_j') \right\|^2, \tag{10}$$

where $N(\cdot, \delta\mathbf{I})$ is Gaussian noise with diagonal covariance matrix, with $\delta$ as the diagonal elements. $\mathcal{H}(\cdot)$ denotes the entropy, and $\hat{\mathbf{y}}_j'$ and $\tilde{\mathbf{y}}_j'$ are the predicted class distributions for node $j$ with and without perturbation, respectively. This metric is based on the principle that more reliable teacher soft labels are more robust to feature perturbations.

During knowledge distillation, given a target node $v$, we sample a set of nodes from its neighbourhood. The sampling distribution is determined by the reliability metric:

$$p(j|\rho_j, \alpha) \propto 1 - \left( \frac{\rho_j}{\rho_{\max}} \right)^\alpha, \tag{11}$$

where $\rho_{\max} = \arg\max_{i \in \mathcal{V}} \rho_i$, and $\alpha$ is a learnable power parameter. To learn $\alpha$, we follow the procedure of (Wu et al., 2023). During training of the overall model, we strive to match the class distribution predicted by the student $\hat{\mathbf{y}}_v$ with the teacher-provided soft labels of the sampled nodes. We therefore include a loss term:

$$Loss_{KRD} = \frac{1-\nu}{|\mathcal{V}|} \sum_{v \in \mathcal{V}} \mathbb{E}_{\substack{u \in \mathcal{N}(v) \\ u \sim \bar{p}(u|\rho_u, \alpha)}} KL(\hat{\mathbf{y}}_v, \hat{\mathbf{y}}_u'), \tag{12}$$

where, for each node $v$, the expectation is with respect to a distribution $\bar{p}(u|\rho_u, \alpha)$, which is proportional to $p(u|\rho_u, \alpha)$ defined in equation 11, for the nodes $u$ in the neighborhood of $v$, and zero elsewhere. The hyperparameter $\nu$ is the same as in equation 9.

Our model consist of $L$ MoE layers (see Figure 3). The training objective is a combination of the cross-entropy loss, the distillation loss, the KRD loss, and embedding losses for every MoE layer of the model. With a model with $L$ MoE layers, we introduce weights $\alpha$, $\beta$, and $\gamma$ to determine the influences of the associated embedding losses:

$$Loss = Loss_{KD} + Loss_{KRD} + \sum_{i=1}^{L} \alpha Loss_{VQ}(i) + \beta Loss_{SS}(i) + \gamma Loss_{LB}(i). \tag{13}$$

### 4.3 MoE initialization

In most implementations, the embeddings of the experts are initialised randomly (Chi et al., 2022; Li et al., 2022; Yan & Li, 2023). In our approach, we desire more informative initialisation, because embeddings are operating in the input space. We therefore apply a pretraining stage, following Zhang et al. (2021a). We pretrain the model for several epochs, routing all input vectors to the first expert. Subsequently, we clone the parameters of the first expert to all other experts. We reset the optimiser state after pretraining. We collect the inputs for each MoE layer and apply $L_2$ normalization. We then apply K-means clustering, with the number of clusters equal to the number of experts. We initialise the embeddings with the cluster centers.

Operating directly in the space of the hidden representations not only removes the superfluous parameters of the projection matrix from the model, but it also makes initialization of the experts much easier. This helps to avoid representation collapse that can be caused by randomly initialized projection matrix.

# 5 Experiments

We evaluate our model on nine real-world datasets. We show that our model can utilize additional parameters more efficiently than a parameter-inflated MLP, an ensemble of MLPs, or a vanilla mixture-of-experts model. We conduct an ablation study to show how the various loss terms influence accuracy.

## 5.1 Experimental setting

**Datasets.** To conduct our experiments we use nine real-world datasets: Cora (Sen et al., 2008), Citeseer (Giles et al., 1998), Pubmed (McCallum et al., 2000), Amazon-Photo, Amazon-Computers, Academic-CS, Academic-Physics (Shchur et al., 2018), OGB-ArXive and OGB-Products (Hu et al., 2020). For the Cora, Citeseer, and Pubmed datasets, we follow the data splitting strategy specified by Kipf & Welling (2016). For the Amazon-Photo, Amazon-Computers, Academic-CS, Academic-Physics, we follow the procedure employed by Zhang et al. (2021b), Tian et al. (2022) and Wu et al. (2023). We randomly split the data into train/val/test subsets. Each random seed corresponds to a different data split. For the OGB-ArXive and OGB-Products we use the public data splits provided by Hu et al. (2020). Dataset statistics are provided in Table 7. For the Amazon-Photo, Amazon-Computers, Academic-CS, Academic-Physics, OGB-ArXive and OGB-Products datasets, we use batched updates due to the large number of nodes and edges.

When presenting and discussing results, we divide the datasets into large, medium and small categories, according to the number of training nodes available. Our method focuses on large and medium-sized datasets. In general, more complicated architectures and distillation procedures struggle to achieve performance gains on small datasets, as demonstrated by Zhang et al. (2021b).

**Baselines.** We compare to three node classification baselines that use GNN-to-MLP knowledge distillation: NOSMOG (Tian et al., 2022), KRD (Wu et al., 2023) and GLNN (Zhang et al., 2021b). All baselines are reproduced using provided official code and hyperparameters.[1] By default, we use GraphSAGE (Hamilton et al., 2017) as the teacher model, in order to facilitate comparison with previous work. We do, however, examine how the method performs with other teacher models. We also compare to CoHOp (Winter et al., 2024), a baseline method that does not employ a teacher, but has a higher inference cost. In order to compare results with a similar parameter count, we provide four parameter-inflated baselines: NOSMOG+, KRD+, GLNN+ (Zhang et al., 2021b) and CoHOp+. For these baselines, we increase the number of student parameters to be 8 times the teacher size, following Zhang et al. (2021b).[2] We conduct 10 runs for our method and each baseline, with the same sequence of seeds for all methods.

**Model.** Our Routing-by-Memory models have the same number of layers as the teacher. Every expert is a linear layer with the same size as the corresponding layer of the teacher. We use up to 8 experts for RbM (the exact number for each dataset is selected using the validation set). We use the same number of experts for all the RbM layers inside the model. Three experts are active at a time. RbM routing is sensitive to dropout, so we avoid the application of dropout directly before the RbM layers. In our model, we apply dropout before the input of the expert for each layer except the first one. We use a linear annealing schedule for the updating of the expert embeddings (see Appendix G).

**Evaluation protocol.** We report the mean and standard deviation of accuracy for ten separate runs with different random seeds. We use the same sequences of seeds reported as Tian et al. (2022) and Wu et al. (2023). We use validation data to select the optimal model. The hyperparameter selection procedure is described in Appendix C. We measure model performance using test data.

We conduct our experiments in two settings: transductive (*trans*) and inductive (*ind*). For the transductive setting we train the model on the full sets of nodes, $\mathcal{V}$, and edges, $\mathcal{E}$. Classification loss is only computed over $X^L$ and $Y^L$, but soft labels for KL-divergence and KRD losses are computed on the full sets of $X$ and $\mathcal{E}$. In the transductive setting we evaluate the model over $X^U$ and $Y^U$. For the inductive setting, we split the unlabeled nodes, $\mathcal{V}^U$, into a set of observed nodes, $\mathcal{V}^U_{obs}$, and a set of inductive nodes, $\mathcal{V}^U_{ind}$, by randomly selecting 20% of the nodes as the inductive subset, following the procedure of Tian et al. (2022)

---

[1]We were not able to reproduce reported NOSMOG results on the OGB-Products dataset for the inductive setting using the official code; for this dataset and setting we provide results using our implementation of NOSMOG.

[2]We are not able to provide results for KRD+ on OGB-ArXive as it requires more than 32 GB of VRAM to run.

and Zhang et al. (2021b). We partition each graph in such a way that the training nodes $\mathcal{V}_{train} = \mathcal{V}^L \sqcup \mathcal{V}^U_{obs}$ and inductive nodes $\mathcal{V}^U_{ind}$ have no edges between them. We denote the edges between the training nodes as $\mathcal{E}_{train}$. Classification loss is only computed over $X^L$ and $Y^L$, but soft labels for KL-divergence and KRD losses are computed over $X_{train} = X^L \sqcup X^U_{obs}$ and $\mathcal{E}_{train}$. In the inductive setting we evaluate the model over $X^U_{ind}$ and $Y^U_{ind}$. For both settings, the embedding losses are computed for all model inputs during training.

In order to achieve the best baseline performance under our evaluation protocol, we reproduce baseline results with publicly available hyperparameter configurations for the datasets. We do not provide results for GLNN, NOSMOG, and CoHop for the Academic-CS and Academic-Phy datasets, nor KRD for the Amazon-Comp dataset, because they are not reported in the original papers. All four algorithms are sensitive to hyperparameter selection; we cannot identify values that lead to reasonable performance on these datasets.

To compare performance across the datasets, we report median *Score*, where Score is a Min-Max normalization of the mean accuracy into the $[0, 1]$ interval. The algorithm with worst performance on the dataset obtains a Score of 0 and the best performing algorithm is assigned a Score of 1. We apply the Skillings-Mack test with a significance level of 5% to assess performance of the algorithms across all datasets. The test is applied separately for each evaluation setting (transductive and inductive).

## 5.2 Performance comparison

We compare our method to GLNN, KRD, NOSMOG and CoHOp baselines. Results are presented in Table 1 for GraphSAGE as the teacher, and in Table 2 for more advanced teacher GNNs. We use RevGNN-Wide (Li et al., 2021) and DRGAT (Zhang et al., 2023) as the advanced teachers, because they are among the best performing GNN models for the larger OGB datasets.

We make the following observations:

1. Table 1 shows that RbM consistently ranks first or second for the medium and large datasets. It can be successfully applied to small datasets but without meaningful performance gains. RbM outperforms all the baselines on medium sized datasets. It is outperformed only by CoHOp on the large datasets. We discuss this in Section 5.4, but for now, we note that CoHOp employs computationally burdensome label propagation.

2. KRD, NOSMOG, and RbM often outperform the teacher model. This has been observed previously (Wu et al., 2023; Tian et al., 2022). Distillation can lead to better generalization and renders the prediction architecture less susceptible to spurious edges. In addition, the students form predictions using both node features and structural information (via positional encoding or DeepWalk), whereas the teacher focuses primarily on the features (the impact of graph structure is much less direct, arising from message passing).

3. Although GLNN performs reasonably well for small and medium datasets, with accuracy close to that of the teacher, it struggles with the large OGB datasets. The node feature information is highly informative for the small/medium datasets. In contrast, for the large datasets, with sparser labelling, the access to graph information is important. This is achieved by neighbourhood aggregation for the teacher, and positional encoding for the students.

4. Table 2 indicates that better teachers lead to improved performance of the distilled models (except for GLNN). The proposed method demonstrates similar outperformance with respect to the baselines. The distilled models do not outperform the more advanced teachers that can more effectively incorporate graph information.

In order to demonstrate that our approach leverages additional parameters better than the baselines, we conduct experiments with parameter-inflated baselines (see Table 3). These expanded baselines are denoted GLNN+, KRD+, NOSMOG+ and CoHOp+. In this experiment every baseline has the number of parameters increased 8 times the teacher size, as in Zhang et al. (2021b) and Tian et al. (2022). The size of every hidden layer is increased, but other parameters are not changed. CoHOp does not use a teacher; we increase the number of parameters by a factor of 8 compared to the model in the original paper. Table 3 indicates that a larger number of parameters can improve performance for some cases, but we do not observe a consistent performance improvement for any baseline. Even when there is performance improvement, RbM remains the best-performing algorithm in 9 out of the 12 settings for medium and large datasets. We apply the Skillings-

Table 1: Performance comparison with GraphSAGE teacher. Results show accuracy (higher is better). The best model is highlighted in **bold**. Second best is underlined. Scores are statistically significant under the Skillings-Mack test with significance level of 5%. For the inductive setting, $p < 0.003$, and for the transductive setting $p < 0.001$. (*) Baseline is reproduced with changes in the official code.

| Dataset | Eval | GraphSAGE | GLNN | KRD | NOSMOG | CoHOp | RbM |
|---|---|---|---|---|---|---|---|
| | | | | **Small datasets** | | | |
| Cora | ind | **82.13±0.50** | 73.28±1.95 | 65.44±16.35 | 81.03±1.96 | 80.38±1.74 | 77.44±1.08 |
| | tran | 82.08±0.63 | 79.20±3.07 | 84.56±0.62 | 81.97±1.70 | 82.99±1.21 | **84.86±0.46** |
| Citeseer | ind | 70.24±0.62 | 70.11±3.04 | 71.74±0.46 | **71.82±3.06** | 71.77±3.37 | 70.96±0.63 |
| | tran | 70.71±1.68 | 70.52±2.18 | 71.07±8.34 | 72.90±1.37 | **75.64±1.69** | 72.98±0.82 |
| PubMed | ind | 77.60±0.48 | 75.05±2.95 | **81.59±0.60** | 75.69±3.08 | 74.84±3.39 | 81.30±0.49 |
| | tran | 77.36±0.45 | 76.65±2.65 | **81.65±0.46** | 77.56±2.35 | 77.22±2.49 | 81.64±0.14 |
| | | | | **Medium size datasets** | | | |
| Amazon-Comp | ind | 82.28±1.01 | 80.46±1.43 | - | 83.14±1.90 | 80.26±2.52 | **85.07±1.66** |
| | tran | 82.34±1.14 | 82.67±1.91 | | 83.05±1.35 | 81.01±1.58 | **85.22±0.85** |
| Amazon-Photo | ind | 92.14±1.08 | 90.45±1.12 | 91.43±0.82 | 92.16±1.12 | 91.85±1.31 | **93.06±0.74** |
| | tran | 91.93±0.96 | 92.44±0.89 | 93.28±0.38 | 93.16±0.93 | 93.06±1.56 | **93.62±1.25** |
| Academic-CS | ind | 89.25±0.78 | - | 92.28±1.13 | - | - | **93.57±0.65** |
| | tran | 89.04±0.37 | | 93.54±0.54 | | | **93.62±0.25** |
| Academic-Phy | ind | 92.73±0.64 | - | 93.85±0.59 | - | - | **94.67±0.27** |
| | tran | 92.78±0.61 | | **94.35±0.36** | | | 94.34±0.26 |
| | | | | **Big datasets** | | | |
| OGB-ArXive | ind | **71.35±0.62** | 59.13±0.55 | 60.84±0.58 | 67.97±0.46 | 71.18±0.47 | 71.31±0.20 |
| | tran | 71.60±0.26 | 64.68±0.20 | 71.64±0.26 | 70.45±0.37 | **72.79±0.09** | 72.48±0.13 |
| OGB-Products | ind | 76.98±0.48 | 60.22±0.30 | - | 77.29±0.71* | **81.68±0.21** | 80.88±0.24 |
| | tran | 77.47±0.27 | 60.34±0.31 | | 77.19±0.41 | **81.67±0.25** | 81.04±0.37 |
| **Med. Score** | ind | 0.4200 | 0.0000 | 0.5773 | 0.7234 | 0.8951 | **0.9967** |
| | tran | 0.1420 | 0.0000 | 0.9470 | 0.4894 | 0.6696 | **0.9980** |

Mack test to assess whether the RbM outperformance is statistically significant. For all experimental settings, p-values are smaller than 0.02, and in most cases are smaller than 0.005, indicating statistical significance.

Since we perform pretraining and clustering in our model initialization, there is some additional computational overhead compared to training parameter-inflated baselines. In our experiments, pretraining does not exceed 7% of the total number of training epochs on average (in our implementation we use early stop, so the number of epochs can vary). The clustering overhead is effectively insignificant. On our setup, it takes 80 seconds to cluster all required embeddings for RbM on OGB-ArXive, one of the larger datasets.

Table 2: Accuracy for advanced teacher models. Performance with the GraphSAGE teacher is provided for reference. OGB-ArXive dataset is used in transductive setting for the experiments. Scores are statistically significant under the Skillings-Mack test with significance level of 5% ($p = 0.0193$)

| Teacher model | Teacher | GLNN | KRD | NOSMOG | RbM |
|---|---|---|---|---|---|
| GraphSAGE | 71.60±0.26 | 64.68±0.20 | 71.64±0.26 | 70.45±0.37 | **72.48±0.13** |
| DRGAT | 73.63±0.07 | 63.51±0.24 | 72.21±0.11 | 70.71±0.12 | **73.10±0.04** |
| RevGNN | 73.98±0.01 | 63.72±0.24 | 72.44±0.14 | 70.75±0.11 | **73.26±0.06** |
| **Median Score** | | 0.0 | 0.9072 | 0.7397 | **1.0** |

## 5.3 Comparing with ensemble and vanilla MoE

To additionally explore whether our approach is an efficient mechanism for exploiting additional parameters, we construct two baselines: a soft-voting ensemble of MLPs and a vanilla MoE. The soft-voting ensemble consists of several MLP students with the same structure, but different random initializations. The three-MLP ensemble has the same inference cost as active experts in the Mixture of Experts, and the eight-MLP ensemble has the same parameter count as all experts. Appendix F provides extensive analysis on parameter number and inference complexity. The vanilla Mixture of Experts model is structured in the same way as the proposed RbM model, but it uses a routing scheme where the policy network and embeddings are initialized randomly and updated with backpropagation (Chi et al., 2022; Li et al., 2022; Yan & Li, 2023). For all

Table 3: Comparison with scaled baselines, with GraphSAGE teacher. Results show accuracy (higher is better). Sizes of the baseline MLPs are adjusted to 8 time the teacher size. RbM uses up to 8 experts. The best model is highlighted in **bold**. Second best is underlined. Scores are statistically significant under the Skillings-Mack test (Skillings & Mack, 1981) with a significance level of 5%. For both transductive and inductive settings, $p < 0.001$.

| Dataset | Eval | GraphSAGE | GLNN+ | KRD+ | NOSMOG+ | CoHOp+ | RbM |
|---|---|---|---|---|---|---|---|
| | | | | **Small datasets** | | | |
| Cora | *ind* | **82.13±0.50** | 73.54±2.18 | 65.50±16.34 | 77.38±5.39 | 80.12±2.30 | 77.44±1.08 |
| | *tran* | 82.08±0.63 | 78.97±3.18 | 84.78±0.61 | 76.04±10.08 | 82.99±1.21 | **84.86±0.46** |
| Citeseer | *ind* | 70.24±0.62 | 67.85±2.79 | **71.83±0.31** | 52.68±7.57 | 71.77±3.37 | 70.96±0.63 |
| | *tran* | 70.71±1.68 | 71.14±2.14 | 70.57±9.58 | 71.24±4.97 | **75.64±1.69** | 72.98±0.82 |
| PubMed | *ind* | 77.60±0.48 | 75.14±2.79 | **81.68±0.43** | 75.30±3.10 | 74.84±3.39 | 81.30±0.49 |
| | *tran* | 77.36±0.45 | 76.13±2.51 | 81.58±0.40 | 77.69±2.41 | 77.22±2.49 | **81.64±0.14** |
| | | | | **Medium size datasets** | | | |
| Amazon-Comp | *ind* | 82.28±1.01 | 79.98±1.21 | - | 83.18±1.65 | 80.27±2.51 | **85.07±1.66** |
| | *tran* | 82.34±1.14 | 82.45±2.14 | | 83.21±1.55 | 81.01±1.58 | **85.22±0.85** |
| Amazon-Photo | *ind* | 92.14±1.08 | 89.92±1.75 | 91.43±0.59 | 92.29±1.13 | 91.85±1.37 | **93.06±0.74** |
| | *tran* | 91.93±0.96 | 92.24±1.32 | 93.36±0.24 | 92.91±1.10 | 93.06±1.56 | **93.62±1.25** |
| Academic-CS | *ind* | 89.25±0.78 | - | 92.57±1.07 | - | - | **93.57±0.65** |
| | *tran* | 89.04±0.37 | | **93.77±0.51** | | | 93.62±0.25 |
| Academic-Phy | *ind* | 92.73±0.64 | - | 94.06±0.55 | - | - | **94.67±0.27** |
| | *tran* | 92.78±0.61 | | **94.50±0.37** | | | 94.34±0.26 |
| | | | | **Big datasets** | | | |
| OGB-ArXive | *ind* | **71.35±0.62** | 60.44±0.69 | OOM | 68.79±0.77 | 65.95±3.35 | 71.31±0.20 |
| | *tran* | 71.60±0.26 | 70.23±0.23 | | 67.37±5.82 | 66.22±5.81 | **72.48±0.13** |
| OGB-Products | *ind* | 76.98±0.48 | 73.83±0.23 | - | 76.30±1.15* | 77.11±0.27 | **80.88±0.24** |
| | *tran* | 77.47±0.27 | 77.17±0.31 | | 77.50±0.44 | 77.95±0.28 | **81.04±0.37** |
| **Med. Score** | *ind* | 0.4519 | 0.0000 | 0.7270 | 0.6287 | 0.5050 | **1.0000** |
| | *tran* | 0.0775 | 0.1834 | 0.9900 | 0.1750 | 0.2016 | **1.0000** |

models, each sample is routed to three experts. We pretrain the vanilla Mixture of Experts model for several epochs, routing all inputs to the first expert. The weights of the pretrained first expert are then cloned to the other experts. Embeddings are not updated during the pretrain stage.

Table 4 shows that RbM outperforms the soft-voting ensemble and vanilla MoE baselines in 10 out of 12 of the settings for the medium and large datasets. The vanilla MoE outperforms the ensemble of MLPs for most settings (8 out of 12) on the medium and large datasets. The vanilla MoE has fewer hyperparameters than RbM, and thus it may be preferable if there is a need to reduce the tuning overhead.

### 5.4  Ablation study, label propagation, and number of experts

**Loss terms.**  We now examine whether each component of the equation 13 is important for achieving better performance. Our model includes three additional loss terms for each RbM layer (see equation 13): commitment loss (6), self-similarity loss (7), and load balance loss (8). In order to conduct the ablation study we remove each component individually. None of the other hyperparameters is changed. The results are presented in Table 5, and show that removing any of the loss components reduces performance. The performance deterioration is relatively small for each dataset, but it is observed for every setting and every loss. It is not clear that any of the three loss terms is the most important. Any combination that includes two loss terms outperforms (by a small margin) the baseline with all three removed for all medium and large datasets, indicating that all three loss terms contribute.

**Label propagation.**  By default, we use DeepWalk positional encoding as an additional set of features. RbM is fully compatible with additional positional information that can be extracted from the graph. As an example, Table 6 demonstrates that including the label propagation information from CoHOp can increase the performance of RbM on the datasets where a considerable portion of the nodes are labeled (OGB-ArXive, OGB-Products). Generating the label propagation information requires propagating and averaging one-hot encoded training labels from a 10-hop neighbourhood around each nodes. In an inductive setting, this must

Table 4: Comparison with ensemble and MoE baselines. 3xMLP and 8xMLP are the soft-voting ensembles of three and eighth MLPs correspondingly trained with a shared teacher, but with different initializations. Vanilla MoE consists of 8 experts, but routes inputs to three. The best model is highlighted in **bold**. Second best is underlined. Scores are statistically significant under the Skillings-Mack test with significance level of 5%. For the inductive setting, $p = 0.004$, and for the transductive setting $p = 0.0337$.

| Dataset | Eval | 3xMLP | 8xMLP | MoE | RbM |
|---|---|---|---|---|---|
| **Small datasets** | | | | | |
| Cora | *ind* | 71.70±2.78 | 66.90±2.43 | 72.98±1.64 | **77.44±1.08** |
| | *tran* | 84.20±0.60 | 84.03±0.25 | **84.86±0.29** | 84.86±0.46 |
| Citeseer | *ind* | 71.60±0.40 | 70.94±0.44 | **71.76±0.47** | 70.96±0.63 |
| | *tran* | 67.57±12.69 | 68.62±11.02 | **73.58±0.43** | 72.98±0.82 |
| PubMed | *ind* | 81.08±0.50 | 80.37±0.43 | **81.58±0.50** | 81.30±0.49 |
| | *tran* | 81.30±0.28 | 81.10±0.29 | **81.80±0.42** | 81.64±0.14 |
| **Medium size datasets** | | | | | |
| Amazon-Comp | *ind* | 84.54±1.51 | 83.87±1.38 | 84.88±1.93 | **85.07±1.66** |
| | *tran* | 84.58±0.86 | 83.72±1.58 | 84.43±0.89 | **85.22±0.85** |
| Amazon-Photo | *ind* | 92.21±1.24 | 92.51±1.43 | 92.66±0.90 | **93.06±0.74** |
| | *tran* | 93.34±1.02 | 93.16±1.03 | 93.56±1.08 | **93.62±1.25** |
| Academic-CS | *ind* | 93.39±0.64 | 93.45±0.66 | 93.43±0.59 | **93.57±0.65** |
| | *tran* | 93.79±0.36 | 93.79±0.37 | **93.97±0.36** | 93.62±0.25 |
| Academic-Phy | *ind* | 94.32±0.45 | 94.11±0.54 | 94.47±0.19 | **94.67±0.27** |
| | *tran* | **94.43±0.32** | 94.34±0.58 | 94.01±0.49 | 94.34±0.26 |
| **Big datasets** | | | | | |
| OGB-ArXive | *ind* | 70.19±0.68 | 69.98±0.52 | 70.96±0.60 | **71.31±0.20** |
| | *tran* | 72.16±0.33 | 72.26±0.32 | 71.49±0.51 | **72.48±0.13** |
| OGB-Products | *ind* | 80.03±0.35 | 80.35±0.38 | 80.80±0.16 | **80.88±0.24** |
| | *tran* | 80.17±0.35 | 80.45±0.34 | 80.64±0.34 | **81.04±0.37** |
| **Median Score** | *ind* | 0.375 | 0.0 | 0.7368 | **1.0** |
| | *tran* | 0.3913 | 0.1747 | 0.8696 | **1.0** |

Table 5: Ablation study on equation 13 loss components. GraphSAGE teacher is used. Results are showing accuracy (higher is better). RbM performance is provided for the reference. For "KD only" column we are reporting results with KD and KDR losses only and no embedding losses.

| Dataset | Eval | RbM (13) | w/o VQ (6) | w/o SS (7) | w/o LB (8) | KD only |
|---|---|---|---|---|---|---|
| Cora | *ind* | 77.44±1.08 | 74.62±1.56 | 75.92±0.89 | 74.68±1.80 | 74.11±1.56 |
| | *tran* | 84.86±0.46 | 84.70±0.46 | 84.64±0.26 | 84.64±0.39 | 84.08±0.64 |
| Citeseer | *ind* | 70.96±0.63 | 69.80±0.20 | 70.22±0.40 | 69.52±0.52 | 70.00±0.30 |
| | *tran* | 72.98±0.82 | 72.38±1.15 | 72.60±0.99 | 72.99±0.91 | 72.28±0.74 |
| PubMed | *ind* | 81.30±0.49 | 80.92±0.29 | 81.00±0.48 | 81.18±0.35 | 80.80±0.95 |
| | *tran* | 81.64±0.14 | 81.38±0.16 | 81.38±0.31 | 81.46±0.36 | 81.28±0.47 |
| Amazon-Comp | *ind* | 85.07±1.66 | 84.22±1.50 | 83.95±1.90 | 84.70±1.65 | 83.92±2.00 |
| | *tran* | 85.22±0.85 | 83.55±1.24 | 84.35±1.10 | 84.02±0.68 | 83.67±0.84 |
| Amazon-Photo | *ind* | 93.06±0.74 | 92.31±1.03 | 92.34±1.03 | 92.76±1.24 | 92.00±1.28 |
| | *tran* | 93.62±1.25 | 93.27±0.96 | 92.98±1.42 | 93.43±1.54 | 92.59±1.41 |
| Academic-CS | *ind* | 93.57±0.65 | 93.46±0.72 | 93.31±0.69 | 93.26±0.72 | 93.10±0.78 |
| | *tran* | 93.62±0.25 | 93.35±0.48 | 93.22±0.50 | 93.12±0.63 | 93.09±0.33 |
| Academic-Phy | *ind* | 94.67±0.27 | 94.62±0.29 | 94.51±0.25 | 94.43±0.27 | 94.35±0.30 |
| | *tran* | 94.34±0.26 | 94.26±0.28 | 94.29±0.22 | 94.25±0.27 | 94.23±0.27 |
| OGB-ArXive | *ind* | 71.31±0.20 | 71.10±0.26 | 71.15±0.29 | 71.28±0.20 | 71.12±0.26 |
| | *tran* | 72.79±0.09 | 72.38±0.21 | 72.43±0.25 | 72.25±0.29 | 72.26±0.27 |
| OGB-Products | *ind* | 80.88±0.24 | 80.54±0.29 | 80.61±0.33 | 80.61±0.31 | 80.38±0.27 |
| | *tran* | 81.04±0.37 | 80.77±0.21 | 80.50±0.42 | 80.78±0.27 | 80.41±0.14 |

be conducted for each new node, and it can become a time bottleneck due to the overhead of fetching node labels from memory.

**Number of experts.** During our experiments we use the same number of experts for all RbM layers in order to reduce the number of hyperparameters. We found that there is an optimal number of experts for RbM for each dataset that can be identified using validation data (from the range $[3, \ldots, 8]$). Appendix D provides results depicting an example of how performance varies as the total number of experts is changed.

Table 6: Applying label propagation positional encoding on OGB datasets.

| Dataset | Eval | w/o Label propagation | | Label propagation | |
|---|---|---|---|---|---|
| | | CoHOp | RbM | CoHOp | RbM |
| OGB-ArXive | *ind* | 54.36±0.79 | 71.31±0.20 | 71.18±0.47 | 75.06±0.45 |
| | *tran* | 54.34±0.72 | 72.79±0.09 | 71.35±0.22 | 72.27±0.21 |
| OGB-Products | *ind* | 33.54±0.61 | 80.88±0.24 | 81.68±0.21 | 81.62±0.25 |
| | *tran* | 32.84±0.57 | 81.04±0.37 | 81.67±0.25 | 81.68±0.25 |

## 5.5 Routing spatial structure analysis

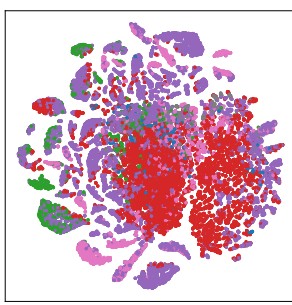 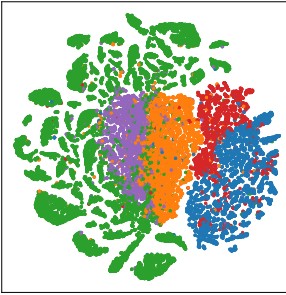 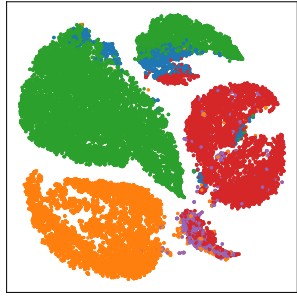 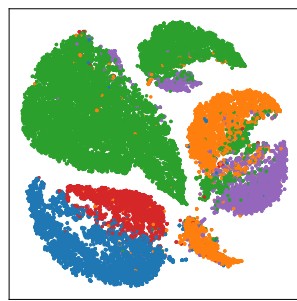

(a) Plain MoE internal representation colored by top expert (left) and by label (right).

(b) RbM internal representation colored by top expert (left) and by label (right).

Figure 4: Analysis of hidden representation for RbM (b) and its projection for MoE (a). Each point represent an instance from the Academic-Physics dataset in transductive setting.

In order to analyse the routing spatial structure qualitatively, we utilise the T-SNE (Van der Maaten & Hinton, 2008), with perplexity of 30 and PCA initialization, to produce a 2-d visualizations of a router embedding space for RbM and a vanilla MoE in Figure 4. These correspond to the hidden representation $h$ for RbM (see equation 3) and the linear projection of the hidden representation, $Wh$, for the MoE (see equation 2). We trained both MoE and RbM models on the Academic-Physics dataset in the transductive setting with the same teacher and selected the router of the last layer to produce the representation. In Figure 4, the hidden representations are colored according to the labels (on the left) and according to the top-score expert (on the right). Figure 4 shows that MoE experts mix and distribute datapoints of the same classes between different experts, while RbM experts have a clear specialization.

## 6 Conclusion and Future work

In this paper we focused on the task of distillation from a graph neural network and introduced RbM, a Mixture of Experts model that encourages strong expert specialization at the routing level. We established how parameter inflation can positively affect the performance and showed practical application of MoE in the knowledge distillation domain. Our approach outperforms existing baselines on most medium-size or large datasets. The key innovations of our approach is in embeddings to be the part of the hidden representation space.

We used additional losses hidden space is forced into the shape of multiple elliptical clusters, which could be too restrictive and thus suboptimal. In order to reduce number of hyperparameres we assumed all layer of RbM to have the same number of experts and thus clustered into the same number of clusters. Selecting a suitable number of experts for the layer automatically can improve the performance of the overall model. Both off these assumptions led to the model being more sensitive to the number of experts than the plain MoE. We leave those direction of improvement as a future work. In addition to that a a future direction can be about the application of the MoE in graph domain. This work uses a concatenation of positional and feature vector for both routing and selected experts processing, however an alternative approach can be to route with positional vector while feature vector is supplied to the selected expert.

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

## A   Datasets description

In Table 7 we provide the key statistics of the datasets we used to evaluate our models: Cora (Sen et al., 2008), Citeseer (Giles et al., 1998), Pubmed (McCallum et al., 2000), Amazon-Photo, Amazon-Computers, Academic-CS, Academic-Physics (Shchur et al., 2018), OGB-ArXive and OGB-Products (Hu et al., 2020).

Table 7: Dataset statistics

| Dataset | # Nodes | # Edges | # Features | # Classes |
|---|---|---|---|---|
| Cora | 2485 | 5069 | 1433 | 7 |
| Citeseer | 2110 | 3668 | 3703 | 6 |
| PubMed | 19717 | 44324 | 500 | 3 |
| Amazon-Comp | 13381 | 245778 | 767 | 10 |
| Amazon-Photo | 7487 | 119043 | 745 | 8 |
| Academic-CS | 18333 | 81894 | 6805 | 15 |
| Academic-Phy | 34493 | 247962 | 8415 | 5 |
| OGB-ArXive | 169343 | 1166243 | 128 | 40 |
| OGB-Products | 2449029 | 61859140 | 100 | 47 |

## B   Hardware specification

Our experiments were conducted using an NVIDIA Tesla V100 GPU with 32GB of memory. The machine has an Intel Xeon Gold 6140 CPU with clock frequency of 2.30GHz and total thread count of 36. All computations, with exception of the clustering, were executed on the GPU. For Cora, Citeseer, PubMed, Amazon-Comp, Amazon-Photo and Academic-CS datasets we executed five parallel runs simultaneously. Each parallel run was allocated 6GB of GPU memory and 5 threads for the clustering. For Academic-Phy, OGB-ArXive, OGB-Products we executed only one run at a time with 32GB of GPU memory and 10 threads for clustering. We were unable to run KRD+ on our setup as it requires more than 32GB of memory.

## C   Hyperparameter tuning

We use Ray Tune (Liaw et al., 2018) to tune model hyperparameters. Specifically, we use the Optuna search algorithm Akiba et al. (2019). We sample 200 hyperparameter configurations for small and medium datasets and 80 for the large datasets. We tuned the following model structure hyperparameters: (i) dropout rate was selected from $[0.0, 0.1, 0.2, 0.3, 0.4, 0.5, 0.6]$ and applied to all dropout layers in the model; (ii) total number of experts was selected from $[4, 5, 6, 7, 8]$. In addition to the structure hyperparameters, we selected the following training hyperparameters: (i) learning rate for Adam optimizer (Kingma & Ba, 2014) was chosen

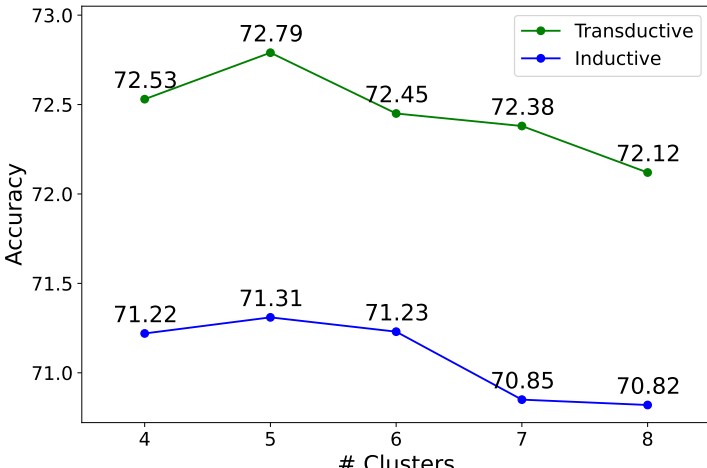

Figure 5: Test set accuracy with respect to the number of experts/clusters for RbM on OGB-ArXive dataset. The optimal number of clusters (5) is clearly identifiable in both transductive and inductive cases.

from $[0.01, 0.005, 0.001]$; (ii) weight $\alpha$ of the commitment loss (6) from the range $[0.0, 0.1]$; (iii) weights $\beta$ and $\gamma$ of the the load-balancing loss (8) and self-similarity loss (7) correspondingly from the range $[0.0, 0.05]$.

Comparing to a distillation setup with MLP student, our training setup introduces three loss hyperparameters (see equation (13)) and an embedding update hyperparameter (see equation (4)). Due to the usage of hyperparameter sampling, these additional hyperparameters do not increase the search time.

## D  Selecting the number of experts

As discussed in the main text, the performance of RbM does vary as the total number of experts is changed. During our experiments we use the same number of experts/clusters for all RbM layers in order to reduce the number of hyperparameters. We found that there is an optimal number of experts for RbM that can be identified for each dataset using the validation dataset (from the range $[3, \ldots, 8]$).

Figure 5 shows how the accuracy depends on the number of experts for the OGB-ArXive dataset. These results are for the test set, but similar results are observed for the validation set, thus allowing selection of the best model. The optimal number of experts can be clearly identified for both the transductive and inductive settings (in the depicted case, 5 for each).

## E  GCN Teacher Model

We investigate whether our model is compatible with an alternative teacher GNN and demonstrates the same advantages over the baselines. The main paper provides results for GraphSAGE as the teacher, together with some results for more advanced GNN teachers. Table 8 provides additional results for experiments in a transductive setting with GCN (Kipf & Welling, 2016) as the teacher. We use the originally reported parameter settings of the baselines for this experiment. These were selected for a GraphSAGE teacher, so it is possible that parameter tuning could improve performance.

## F  Complexity analysis

In this section we characterize the complexity of the models. All the layers in each model are identically structured, and the number of layers is the same as the teacher's number of layers. Thus, we now characterize

Table 8: Distillation to RbM results using GCN teacher in transductive setting. Results are showing accuracy (higher is better). The best model is highlighted in **bold**. Second best is underlined. Scores are statistically significant under the Skillings-Mack test with significance level of 5% ($p = 0.0015$)

| Dataset | GLNN | KRD | NOSMOG | RbM |
|---|---|---|---|---|
| Cora | 79.39±1.64 | 84.42±0.57 | 80.93±1.65 | **85.02±0.29** |
| Citeseer | 69.28±3.12 | **74.86±0.58** | 73.78±1.54 | 74.24±0.20 |
| PubMed | 74.81±2.39 | 81.98±0.41 | 75.80±3.06 | **82.74±0.09** |
| Amazon-Comp | 82.63±1.40 | - | 83.72±1.44 | **84.41±1.56** |
| Amazon-Photo | 92.68±0.56 | 92.21±1.44 | 92.44±0.51 | **93.61±0.91** |
| Academic-CS | - | **94.08±0.34** | - | 93.10±0.40 |
| Academic-Phy | - | 94.30±0.46 | - | **94.31±0.15** |
| OGB-ArXive | 61.46±0.33 | 70.92±0.21 | 71.10±0.34 | **71.67±0.19** |
| OGB-Products | 63.92±0.61 | - | 77.41±0.21 | **79.20±0.15** |
| **Median Score** | 0.0 | 0.9042 | 0.6124 | **1.0** |

Table 9: Complexity analysis for the student models. Since we are using sparse routing, we present active parameter counter along with the total parameters. Only active parameters are participating in the inference. Note that these empirical results are specific to our setup and are not generalised.

| Layer | Parameter count | | Inference time complexity |
|---|---|---|---|
| | Total | Active | |
| MLP | $F(N+1)$ | $F(N+1)$ | $O(FN)$ |
| Ensemble | $E^a F(N+1)$ | $E^a F(N+1)$ | $O(E^a FN)$ |
| MoE | $F(EN+E+1)+H(F+E)$ | $F(E^a N+E^a+1)+H(F+E)$ | $O(E^a FN+EH+FH)$ |
| RbM | $EF(N+1)+EF$ | $E^a F(N+1)+EF$ | $O(E^a FN+EF)$ |

the parameter count and computational complexity of a single layer for MLP, MoE and RbM. We also contrast this with an ensemble of MLPs. The feature size of the input vector is denoted by $F$. We assume that MLP layer is a linear layer of projection size $N$ and a bias. We denote the total number of experts in MoE and RbM by $E$, and the number of active experts during inference by $E^a \leq E$. Note that $E \ll F$ and $E \ll N$. We set the number of MLPs in the ensemble to be equal to the number of active experts. Each expert of MoE or RbM layer is a linear layer of projection size $N$ and a bias. Routing is conducted according to equation 3 for RbM and equation 2 for the MoE. During the inference, routing procedure is conducted to descide which experts to run, therefore active parameter counter and time complexity both contain $E$ dimensionality. MoE routing embeddings have internal size of $H$. Note that $H \gg E$ and it is common for intermediate MoE layers to have $H = N$. For the RbM layer we utilise a trainable constant attention that multiplies each feature with a scalar. In the ensemble input is always routed to all the members, thus all parameters of the ensemble are contributing to computation complexity.

From Table 9 one can see that MoE and RbM have some additional complexity comparing to the ensemble of MLPs that comes from the router. Thus if ensemble is expected to be $E^a$ times slower that MLP student, RbM is expected to be $E^a + 1$ times slower. However, RbM is faster than MoE that utilises projection into embedding space while RbM uses embeddings that are in input space.

## G   Annealing schedule

In order to ensure the convergence of expert embeddings, we apply a linear annealing schedule:

$$\lambda(t) = \lambda_0 + \frac{(1-\lambda_0)\Delta}{T}t, \tag{14}$$

where $0 \leq \Delta < 1$ is an annealing hyperparameter, $T$ is the expected number of epochs and $\lambda_0$ is a constant that specifies the initial update rate. In our experiments, we set $\lambda_0 = 0.9$, $T = 200$ and $\Delta = 0.05$.

With the following experiment, we demonstrate how varying $\Delta$ affects performance of RbM. We repeat the inductive and transductive experiments for the OGB-ArXive dataset using GraphSAGE as a teacher model,

Table 10: Routing-by-Memory with annealing.

| Dataset | Eval | $\Delta = 0.0$ | $\Delta = 0.05$ | $\Delta = 0.1$ | $\Delta = 0.5$ | $\Delta = 0.9$ |
|---------|------|----------------|-----------------|----------------|----------------|----------------|
| OGB-ArXive | *ind* | 71.31±0.20 | 71.31±0.20 | 71.27±0.24 | 71.26±0.27 | 71.27±0.42 |
| | *tran* | 72.48±0.13 | 72.48±0.13 | 72.46±0.22 | 72.44±0.20 | 72.34±0.25 |

but vary the annealing parameter $\Delta$. From Table 10, we can observe how for larger values, there is a minor performance drop, but generally RbM is relatively insensitive to the annealing schedule. We assume that large values of $\Delta$ potentially lead to the algorithm becoming stuck at poorer expert embeddings. Empirically, we observe that, even with $\Delta = 0$, the expert embeddings converge sufficiently such that all representations are consistently routed to exactly the same experts in successive epochs.

