# OpenReview forum: "Graph Knowledge Distillation to Mixture of Experts"
_TMLR — Accepted by TMLR_

### Review · Reviewer_qpbM · 2024-07-24

**Summary Of Contributions:**

This paper proposes a novel knowledge distillation approach from Graph Neural Networks (GNNs) to Multi-Layer Perceptrons (MLPs) to address latency issues in GNNs. The introduced model, Routing-by-Memory (RbM), is a Mixture-of-Experts (MoE) architecture that enhances expert specialization. By encouraging each expert to focus on specific regions in the hidden representation space, the RbM model achieves consistent performance across multiple datasets. Experimental results demonstrate that RbM outperforms traditional MLPs and other baseline models, providing a scalable solution for node classification tasks in large-scale graph-based applications.

**Audience:**

Yes

**Claims And Evidence:**

Yes

**Requested Changes:**

as listed in weakness

**Strengths And Weaknesses:**

**Strengths**
1. Better Performance: The Routing-by-Memory (RbM) model demonstrates superior performance over traditional MLPs and baseline models across multiple datasets.

**Weaknesses**
1. Lack of Novelty: The approach seems a direct application of Mixture-of-Experts (MoE), limiting its novelty.
2. Unclear Motivation: The specific motivation for using MoE in graph knowledge distillation is not well explained.
3. Model Complexity: The added complexity of the RbM model requires more effort in tuning hyperparameters.
4. Pretraining Overhead: Requires a pretraining phase for expert embeddings, adding computational overhead.

---

> ### Author Response · Authors · 2024-08-31
> **Official response to Reviewer qpbM (1/3)**
>
> Thank you for your review of our work. We would like to address your concerns.
>
> **W1. Novelty**: We clearly stated the novelty of the work in the last two paragraphs of the introduction section, but we are happy to clarify. We have revised the introduction section with the following lines to, hopefully, make it even clearer.
>
> 1. Ours is the first work to propose the use of a student Mixture-of-Experts (MoE) model for Distillation of GNNs.
> 1. We propose a novel MoE model that differs from, and outperforms, a standard sparse MoE. In particular, we introduce a novel routing system. Although this routing system was previously proposed for routing to a single expert in a computer vision setting, we introduce important adaptations, allowing for routing to multiple experts and using a different distance.
> 1.  During training of the proposed MoE, we introduce several loss terms to encourage better clustering of representations and improved expert specialization.
> 1. We conduct multiple novel experiments, demonstrating that the proposed approach consistently outperforms both (i) enlarged MLP students; and (ii) ensembles or sparse MoEs.

---

> > ### Author Response · Authors · 2024-08-31
> > **Official response to Reviewer qpbM (2/3)**
> >
> > **W2. Motivation**:
> > The motivation was explained at the end of the introduction, in light of the reviewer's observation, we have revised it to make it clearer. Please see the blue text in the first paragraph and the last two paragraphs of Section 1 of the revised paper.  In case the reviewer has any further constructive suggestions regarding which aspects of the motivation are unclear, we would be very happy to modify the text.
> >
> > In this response, we think it is useful to provide a summary discussion.
> >
> > First, the motivation for GNN distillation, in general, is as follows. There are multiple examples of very large GNNs being deployed in industrial settings (e.g., Pinterest, fraud detection systems, app-store recommender systems). The features associated with the nodes in these GNNs are updated often. For example, in an app-store recommender setting, the features of a node (user) are updated every time a user interacts with the app store. Item features are updated at least daily based on ordering, popularity, ratings, etc. Due to this evolution of features, at inference time, when the GNN must compute an output for the node, it must retrieve the features of neighbouring nodes for the GNN aggregation step. For a GNN with multiple layers, the receptive field can be very large. In standard GNNs, each layer extends the aggregation neighbourhood by another hop, and a 4-hop or 5-hop neighbourhood can span hundreds of nodes. The retrieval of the features of these neighbouring nodes can be by far the greatest contributor to latency in response to a query about the status of a node. GNN distillation aims to eliminate the need for aggregation. Ideally, by training a local model, that does not use the features of neighbours, we can greatly reduce the response latency. Indeed, experiments show that an MLP student responds 10x or even 100x faster than a teacher GraphSAGE model.
> >
> > Now we turn to the motivation of our work. Typically, in previous work on GNN distillation, the student Multi-Layer Perceptron (MLP) is chosen to be the same size (number of parameters) as the teacher GNN. However, previous works (NOSMOG and GLNN) have experimentally demonstrated that it may be beneficial for performance to increase student size by a factor of four or eight, especially for larger datasets. Our experiments, reported in the paper, show that this improvement is very inconsistent - sometimes a larger model performs considerably better, but other times the performance is no better, or even worse (please compare Tables 1 and 3 - for example, for OGB-ArXive and OGB-PRoducts, there is a dramatic improvement for GLNN, but NOSMOG does not improve, and GLNN is still worse than NOSMOG). This motivates us to consider how to use the extra parameters more effectively, to consistently improve performance without dramatically increasing the latency or computation overhead during inference.
> >
> > One immediate option is a simple ensemble of MLPs, and this approach does offer performance improvement. But if we have 8 times the number of parameters, the inference computational cost is increased by a factor of 8.
> >
> > As an alternative, we might hope to achieve the same performance benefit by routing each sample to a subset of the MLPS. This strategy can be improved if we perform the routing at each layer or block of layers. If we train such a sparse mixture-of-experts carefully, then we can achieve the same performance improvement as the ensemble, but for only (slightly more than) 3x the inference cost of the MLP.
> >
> > In observing how a standard sparse Mixture-of-Experts behaves, we see that there is unsatisfactory expert specialization - please see Figure 4(a). Experts are responsible for making decisions about many classes. Beyond this, although the hidden representations do exhibit some clustering behaviour, it is unsatisfactory, with classes being separated into multiple small cliques, and limited separation between different classes.
> >
> > We were thus motivated to develop an improved model and training framework. First, we adapted the Routing-by-Memory strategy to our setting, by allowing for routing to multiple experts and changing the distance function. Second, we incorporated additional loss terms to (i) encourage separation of expert embeddings; and (ii) encourage representations to migrate closer to a single expert embedding. These strategies lead to much better expert specialization and improved clustering of class representations - see Figure 4(b).

---

> ### Author Response · Authors · 2024-08-31
> **Official response to Reviewer qpbM (3/3)**
>
> **W3. Model Complexity - Extra Hyperparameters**: It is correct that our proposed model has some additional hyperparameters. We agree that this may require slightly more effort when performing hyperparameter tuning.
>
> We have addressed this hyperparameter tuning issue in the appendixes C and D, where we discuss how we performed tuning automatically with Ray Tune.
> This way, we can benefit from efficient built-in search algorithms like Bayesian Optimization, ZOOpt, or Optuna (which we used). In our setup, the total search for hyperparameters for one model (baseline MLP or proposed Routing-by-Memory) takes 16 hours.
> Due to the usage of hyperparameter sampling instead of a grid search, this number depends only of the maximum number of samples and not on the number of hyperparameters.
> Therefore, the additional effort required for tuning the RbM model is comparison to the MoE model is writing exactly two extra lines of code for the tuning configuration, while tuning takes exactly the same amount of time.
>
> We have added a paragraph in the revised paper in appendix C to acknowledge the extra hyperparameters, and to explain that this induces minimal additional overhead in terms of effort or time.
>
> **W4. Pretraining Overhead**: Our approach does add a minor overhead  due to having a pretraining phase.
> However, we would like to point out that the number of pretraining epochs for RbM does not exceed 7\% of the total number of training epochs on average (in our implementation we use early stop).
>
> The clustering overhead is effectively insignificant.
> On our setup, it takes 80 seconds to cluster all required embeddings for RbM on OGB-ArXive (one of the largest graphs in our experiments).
> We used the SciKit-Learn implementation of K-Means in our experiments for fast prototyping and never deeply optimised our implementation.
> Therefore, a good approach to minimise the overhead is by using a GPU-based implementation of K-Means can be applied to speed-up the process up to 30 times.
> In addition to that reducing the number of training epochs by the number of pretraining epochs will nullifying the pretraining overhead completely.
>
> We have added a paragraph to the Section 5.2 of the revised paper to clarify that there is an additional overhead associated with pre-training, but it is less than 8 percent of the total training time for all of our experiments.
>
>
> **References**
> 1. Shichang Zhang, Yozen Liu, Yizhou Sun, and Neil Shah. Graph-less Neural Networks: Teaching Old MLPs New Tricks Via Distillation. *ICLR*, 2021.
> 1. Yijun Tian, Chuxu Zhang, Zhichun Guo, Xiangliang Zhang, and Nitesh Chawla. NOSMOG: Learning Noise-robust and Structure-aware MLPs on Graphs. *NeurIPS*, 2022.
> 1. Lirong Wu, Haitao Lin, Yufei Huang, and Stan Z. Li. Quantifying the Knowledge in GNNs for Reliable Distillation into MLPs. *ICML*, 2023.
> 1. Richard Liaw, Eric Liang, Robert Nishihara, Philipp Moritz, Joseph E Gonzalez, and Ion Stoica. Tune: A Research Platform for Distributed Model Selection and Training. *arXiv preprint arXiv:1807.05118*, 2018.
> 1. Falkner, Stefan, Aaron Klein, and Frank Hutter. BOHB: Robust and Efficient Hyperparameter Optimization at Scale. *ICML*, 2018.
> 1. Takuya Akiba, Shotaro Sano, Toshihiko Yanase, Takeru Ohta, and Masanori Koyama. Optuna: A Next-generation Hyperparameter Optimization Framework. *KDD*, 2019.
> 1. Yu-Ren Liu, Yi-Qi Hu, Hong Qian, Chao Qian, Yang Yu. ZOOpt: Toolbox for Derivative-Free Optimization. *SCIENCE CHINA Information Sciences*, 2022.
> 1. "Yinyang" K-Means and K-NN using NVIDIA CUDA, 2019, *https://github.com/src-d/kmcuda*.

---

### Review · Reviewer_x9kc · 2024-08-12

**Summary Of Contributions:**

The paper proposes an efficient approach to knowledge disillation of Graph Neural Networks (GNNs) using a specially-designed student model, Routing-by-Memory (RbM), which is a form of Mixture-of-Experts (MoE). The authors provide experimental evidence that simply increasing the size of a student MLP does not consistently lead to better performance. To address this issue, the proposed RbM model encourages expert specialization, leading to more consistent performance across multiple datasets.

**Audience:**

Yes

**Claims And Evidence:**

Yes

**Requested Changes:**

- The proposed MoE, RbM, represent each expert based on a centroid embedding which is estimated with the moving average. This does not seem to be principled and the convergence property of the learning algorithm is not addressed.

- The motivation for the spatial design of the RbM is not well-justified. For example, the RbM can be viewed as a special instance of the sparse MoE while the benefits for the RbM simplification is not discussed.

- The technical contribution of the paper seems incremental. This could be strengthened by taking into account other spatial space beyond the Euclidean.

**Strengths And Weaknesses:**

- Adapting a mixture-of-experts (MoE) model for knowledge distillation (KD) of GNNs is intuitive and could lead to improved performance over the MLP approach.

- The authors propse a modified top-K sparse MoE architecture, called RbM, for the KD task, enforcing each expert to focus on a spatiatial regime in the input embedding space based on the cosine similarity. The RbM model encourages expert specialization, leading to more consistent performance across multiple datasets.

- Empirical evaluation on several real-life datasets demonstrate that RbM can achieve improved accuracy on larger datasets compared to other popular methods as well as the MoE.

---

> ### Author Response · Authors · 2024-08-31
> **Official response to Reviewer x9kc (1/2)**
>
> Thank you for your thoughtful review of the paper.
>
> **RC1.**
> In the original submission, we incorporated the moving average procedure, following the technique proposed by Zhang et al. (2021), with the goal of incorporating a simple clustering update. The update is analogious to the cluster center update in mini-batch spherical K-means.
>
> Reflecting on the convergence question made us reconsider the embedding update design.
> We have now included an annealing schedule for the coefficient $\lambda$ in the equation (4).
> We apply schedule such that $\lambda(t) < \lambda(t+1)$.
> With this change, as $t\rightarrow \infty$, the change to the expert embeddings tends to zero. Provided any annealing in the gradient descent is slower (or adaptive), this approach should guarantee convergence. A formal proof is beyond the scope of this paper.
>
> We have modified the results of the paper using a linear schedule:
> \begin{equation}
>     \lambda(t) = \lambda_0 + \frac{(1-\lambda_0) \Delta}{T} t,
> \end{equation}
> where $0 \le \Delta < 1$ is an annealing hyperparameter, $T$ is a expected number of epochs and $\lambda_0$ is the constant used for experiments in original version of the paper.
> For the following experiments, we use $T = 200$ and vary $\Delta$.
> For a small value, $\Delta = 0.05$, RbM with annealing provides the same results as the version in the original paper.
> For larger values, there is a very minor performance drop.
> This is, in part, due to the hyperparameter optimization procedure being performed for the fixed $\lambda$. Beyond this, a larger value of $\Delta$ potentially leads to the algorithm becoming stuck at poorer expert embeddings.
> Empirically, we observe that the expert embeddings converge sufficiently for practical purposes (i.e., the expert embeddings vary very little and routing choices remain consistent over multiple epochs) even with $\Delta =0$.
>
> **Routing-by-Memory with momentum annealing. Routing-by-Memory results from the paper are provided for reference.**
>
> | **Dataset**                 | **Eval**      |  **RbM**        | **$\Delta = 0.05$** | **$\Delta = 0.1$** | **$\Delta = 0.5$** | **$\Delta = 0.9$** |
> |-----------------------------|---------------|----------------|-----------------|----------------|----------------|----------------|
> | OGB-ArXive | *ind*    | 71.31$\pm$0.20 | 71.31$\pm$0.20  | 71.27$\pm$0.24 | 71.26$\pm$0.27 | 71.27$\pm$0.42 |
> | OGB-ArXive     | *tran* | 72.48$\pm$0.13 | 72.48$\pm$0.13  | 72.46$\pm$0.22 | 72.44$\pm$0.20 | 72.34$\pm$0.25 |
>
>
> As an alternative, we also tested a gradient update of the expert embeddings. To do this, we remove the stop-gradients in equation (3) in the paper, and set:
> \begin{equation}
>     G_{RbMGrad}(h) = \text{softmax} \left( \text{top}_k\left( \frac{Q^{RbM}h}{\|Q^{RbM}\|\|h\|}\right) \right).
> \end{equation}
> Since this embedding update uses gradient descent, the convergence results of Sparse Mixture-of-Experts apply.
> We report the performance of using gradient descent in the Table below.
> We see a slight deterioration.
> This may be partially attributed to the automatic hyperparameter optimization procedure being applied using the moving average procedure.
> In the revised paper, we retain the annealed moving average procedure, because we have not had time to repeat all experiments with gradient descent.
> We have updated equation (4) accordingly.
> Annealing schedule description is added as appendix G.
>
> **Routing-by-Memory with gradient update of embeddings labeled as RbM(grad). NOSMOG and Routing-by-Memory from the paper are provided for reference.**
>
> | **Dataset**  | **Eval** | **NOSMOG**     | **RbM**        | **RbM(grad)**  |
> |--------------|----------|----------------|----------------|----------------|
> | OGB-ArXive   | ind      | 67.97$\pm$0.46 | 71.31$\pm$0.20 | 70.91$\pm$0.74 |
> | OGB-ArXive   | tran     | 72.79$\pm$0.09 | 72.48$\pm$0.13 | 71.83$\pm$0.14 |
> | OGB-Products | ind      | 77.29$\pm$0.71 | 80.88$\pm$0.24 | 80.75$\pm$0.35 |
> | OGB-Products | tran     | 77.19$\pm$0.41 | 81.04$\pm$0.37 | 80.71$\pm$0.43 |

---

> > ### Author Response · Authors · 2024-08-31
> > **Official response to Reviewer x9kc (2/2)**
> >
> > **RC2.**
> > Our motivation for the spatial design of RbM routing is to achieve better clusterings of embeddings, which leads to improved expert specialization, and better performance (see Section 4.3 of our paper). Rather than apply an additional projection, we operate directly in the space of the hidden representations. Including a projection matrix not only adds extra parameters that are unnecessary, but it also makes it more challenging to encourage clustering of the embeddings and the desired expert specialization.
> > We added a paragraph to Section 4.3 to clarify it.
> >
> > RbM is indeed an instance of a sparse MoE with a special routing design.
> > We have emphasised that in Section 4.1.
> > The main benefits of our approach are (i) fewer parameters for the RbM routing compared to typical sparse MoE (see Appendix F and Table 9); (ii) better expert specialization and node embedding clustering (Section 4.3), which we conjecture leads to (iii) better overall performance on larger datasets.
> >
> > **RC3.** We have specified the technical contribution of the paper. To the best of our knowledge, ours is the first work to use a mixture-of-experts for distillation of a graph neural network. We adapt a routing approach from image processing, extending it to allow routing to multiple experts and employing a different distance. In the context of graph learning, this is a novel application of this technique. Our claim is that it improves the clustering of internal embeddings, leading to better expert specialization, and overall accuracy improvement. Our experiments show that the proposed architecture consistently improves over existing distillation methods, ensembles, and vanilla designs of existing Mixtures-of-Experts. Whether the technical contribution is incremental is a matter of opinion, but we would argue that the work represents an advance in the practically important field of distillation methods for graph neural networks, and that our experimental analysis provides sufficient support for the claims of the paper.
> >
> > Conducting computations in another space is very intriguing - thank you for the suggestion. In particular, there is some evidence that hyperbolic geometry can be beneficial in representing complex networks. We would be very interested in exploring a hyperbolic architecture, using hyperbolic MLPs as the experts and making routing decisions based on distances in a Lorentz model. While this is an interesting research direction, it is not feasible to accomplish in a revision of the work. We do not consider that such an extension is necessary to make our technical contribution meaningful and our results useful and interesting for the research community.
> >
> > **References**
> > 1. Noam Shazeer, Azalia Mirhoseini, Krzysztof Maziarz, Andy Davis, Quoc Le, Geoffrey Hinton, and Jeff Dean.
> > Outrageously Large Neural Networks: The Sparsely-Gated Mixture-of-Experts Layer. *ICLR*, 2016.
> > 1. Zewen Chi, Li Dong, Shaohan Huang, Damai Dai, Shuming Ma, Barun Patra, Saksham Singhal, Payal
> > Bajaj, Xia Song, Xian-Ling Mao, Heyan Huang, and Furu Wei. On the Representation Collapse of Sparse
> > Mixture of Experts. *NeurIPS*, 2022.
> > 1. Bo Li, Yifei Shen, Jingkang Yang, Yezhen Wang, Jiawei Ren, Tong Che, Jun Zhang, and Ziwei Liu. Sparse
> > Mixture-of-Experts are Domain Generalizable Learners. *ICLR*, 2022.
> > 1. Roller Stephen, Sainbayar Sukhbaatar, and Jason Weston. Hash Layers for Large Sparse Models. *NeurIPS*, 2021.
> > 1. Taher M Ghazal, Muhammad Zahid Hussain, Raed A Said, Afrozah Nadeem, Mohammad Kamrul Hasan, Munir Ahmad, Muhammad Adnan Khan and Muhammad Tahir Naseem. Performances of K-Means Clustering Algorithm with Different Distance Metrics. *Intelligent Automation & Soft Computing*, 2021.

---

### Review · Reviewer_SUMT · 2024-08-17

**Summary Of Contributions:**

The paper proposes a knowledge distillation for graph neural networks (GNNs) for graph node label prediction. The authors claim that usual MLP does not have good performance even when the network is sufficiently large. Instead, the proposed student model consists of a mixture of expert model for which the authors propose an adaptive weighting strategy called routing by memory. The loss function is designed by combining knowledge distillation loss and embedding loss terms. The knowledge distillation loss has three terms such as cross-entropy with true label, KL divergence with teacher label, and knowledge-aware reliable distillation (KRD) terms (Wu et al., 2023). The embedding loss also has three terms such as the vector quantization (VQ) loss, self-similarity loss, and balance loss (Shazeer et al., 2016). The performance is verified through by using several benchmark datasets.

**Audience:**

Yes

**Claims And Evidence:**

Yes

**Requested Changes:**

- Could you clarify which parts of the proposed method are specifically tailored to graphs? Currently, based on my understanding, it seems that only the generation of node features and position features is graph-specific. The other main components, such as the loss definition and MoE, don't appear to have specific aspects related to graphs. It might be easier for readers to grasp the concept if it were clearer what exactly makes this method 'Graph Knowledge Distillation.'

- When I first read the paper, I mistakenly thought that routing by memory (RbM) was an original idea introduced in this paper. However, the concept comes from Zhang et al. (2021a). I think this should be made clearer, especially since the second point in the list at the end of the introduction is misleading. Further, the descriptions in related work is not sufficiently clear about this issue.

- I think the authors should elaborate on the following point further. Just by looking at Equation (10), it's not clear that the sampling focuses on the important points.
> we sample nodes from its neighbourhood, N(v), updating the sampling distribution after every training step based on how well the student model is predicting each node

- The entire optimization process is guaranteed to converge? The proposed method is not a standard gradient descent, whose convergence is widely studied, because the moving average type update (4) is included in the procedure. Clarifying it would be informative for readers.

- Why increasing the number of parameters of MLP does not consistently lead to better performance?

- What is the necessity of distillation in graph node classification? If the input is a single graph and the task is to predict the node labels, all node predictions have already been assigned by the teacher model once its training is complete. In this case, what is the significance of preparing a separate, lightweight student model? In the inductive setting, the authors mention using 20\% of the labels as test labels—are you assuming a scenario where a much larger amount of test labels is available in practice? Or are you considering a situation where the same model will be applied to a different graph? Alternatively, is the purpose to improve accuracy? I believe clarifying these points would make the usefulness of the approach more apparent.

Minor:

- In (2), superscript Q^MOE is used, while Q_MOE is used before that.

- After (3), 'v'.

- Before (10), ``... node. and include ...''

- Fig 2 and 3 are not referred from the main text.

**Strengths And Weaknesses:**

S: It seems clear that knowledge distillation for GNNs is a topic of significant interest to the community.

S: The paper is easy to follow.

S: The evaluation experiments seem to have been conducted comprehensively.

W: The explanation of the novelty is somewhat unclear (see requested changes).

W: Convergence of the learning process is not clarified.

---

> ### Author Response · Authors · 2024-08-31
> **Official response to Reviewer SUMT (1/3)**
>
> We thank you for your very careful evaluation of our paper, especially for the insightful comments and the identification of minor mistakes.
>
> **Q1.** We would like to point out that our paper is called "Graph Knowledge Distillation" because the pretrained GNN model (teacher) is used as a teacher to distill knowledge into a simpler non-GNN model (student).
> The non-GNN model (MLP in Zhang et al. (2021b), MoE and RbM in our work) is a general-purpose model and is not specially designed to process graphs.
> Unlike GNNs, the student model does not employ node aggregation, which results in a lightweight inference and suitable for cases when node fetching is a performance bottleneck.
>
> The distillation procedure training procedure is, however, graph-specific.
> This is most evident in loss functions (9) and (12) (in the revised paper). In (9), we strive to minimize the distance between the teacher and student node embeddings (cross-entropy for labelled nodes and KL-distance for unlabelled nodes). In (12), we strive to minimize the divergence between the student node representation and those of "reliable" neighbours (nodes whose teacher predictions are robust to feature perturbation). The loss term (12) is relying on the graph structure and implicitly assumes homogeneity in the neighbourhood. As you mentioned, we also use positional encoding features from Tian et al. (2022), that are graph-specific.
>
> **Q2.** We certainly did not intend to give this impression. RbM was presented previously. The major modification is that we allow routing to multiple experts (cf. routing ((4) in Zhang et al. (2021a) and (3) in our paper) and embedding update ((9) in Zhang et al. (2021a) and (4) in our paper).
> This makes our model more general and provides an explicit parallel with typical MoE routing (Shazeer et al., 2016). By setting the number of active experts to one (top-1) routing, our model becomes equivalent to Zhang et al. (2021a), albeit with a different distance.
>
> In order to further emphasize that the routing technique is an adaptation of  Zhang et al. (2021a) and not an original contribution we have added two sentences to introduction. We have explained the difference with Zhang et al. (2021a) with an additional sentence below equation (5).
>
> **Q3.**
> To elaborate upon the sampling procedure, we have added a brief description to the end of the Section 4.2.

---

> > ### Author Response · Authors · 2024-08-31
> > **Official response to Reviewer SUMT (2/3)**
> >
> > **Q4.** Thank you for pointing this out. In conjunction with a similar question by another reviewer, your question made us reconsider the embedding update design.
> > We have now included an annealing schedule for the coefficient $\lambda$ in the equation (4).
> > We apply schedule such that $\lambda(t) < \lambda(t+1)$.
> > With this change, as $t\rightarrow \infty$, the change to the expert embeddings tends to zero. Provided any annealing in the gradient descent is slower (or adaptive), this approach should guarantee convergence. A formal proof is beyond the scope of this paper.
> >
> > We have modified the results of the paper using a linear schedule:
> > \begin{equation}
> >     \lambda(t) = \lambda_0 + \frac{(1-\lambda_0) \Delta}{T} t,
> > \end{equation}
> > where $0 \le \Delta < 1$ is an annealing hyperparameter, $T$ is a expected number of epochs and $\lambda_0$ is the constant used for experiments in original version of the paper.
> > For the following experiments, we use $T = 200$ and vary $\Delta$.
> > For a small value, $\Delta = 0.05$, RbM with annealing provides the same results as the version in the original paper.
> > For larger values, there is a very minor performance drop.
> > This is, in part, due to the hyperparameter optimization procedure being performed for the fixed $\lambda$. Beyond this, a larger value of $\Delta$ potentially leads to the algorithm becoming stuck at poorer expert embeddings.
> > Empirically, we observe that the expert embeddings converge sufficiently for practical purposes (i.e., the expert embeddings vary very little and routing choices remain consistent over multiple epochs) even with $\Delta =0$.
> >
> > **Routing-by-Memory with momentum annealing. Routing-by-Memory results from the paper are provided for reference.**
> >
> > | **Dataset**                 | **Eval**      |  **RbM**        | **$\Delta = 0.05$** | **$\Delta = 0.1$** | **$\Delta = 0.5$** | **$\Delta = 0.9$** |
> > |-----------------------------|---------------|----------------|-----------------|----------------|----------------|----------------|
> > | OGB-ArXive | *ind*    | 71.31$\pm$0.20 | 71.31$\pm$0.20  | 71.27$\pm$0.24 | 71.26$\pm$0.27 | 71.27$\pm$0.42 |
> > | OGB-ArXive     | *tran* | 72.48$\pm$0.13 | 72.48$\pm$0.13  | 72.46$\pm$0.22 | 72.44$\pm$0.20 | 72.34$\pm$0.25 |
> >
> > As an alternative, we also tested a gradient update of the expert embeddings. To do this, we remove the stop-gradients in equation (3) in the paper, and set:
> > \begin{equation}
> >     G_{RbMGrad}(h) = \text{softmax} \left( \text{top}_k\left( \frac{Q^{RbM}h}{\|Q^{RbM}\|\|h\|}\right) \right).
> > \end{equation}
> > Since this embedding update uses gradient descent, the convergence results of Sparse Mixture-of-Experts apply.
> > We report the performance of using gradient descent in the Table below.
> > We see a slight deterioration.
> > This may be partially attributed to the automatic hyperparameter optimization procedure being applied using the moving average procedure.
> > In the revised paper, we retain the annealed moving average procedure, because we have not had time to repeat all experiments with gradient descent.
> > We have updated equation (4) accordingly.
> > Annealing schedule description is added as appendix G.
> >
> > **Routing-by-Memory with gradient update of embeddings labeled as RbM(grad). NOSMOG and Routing-by-Memory from the paper are provided for reference.**
> >
> > | **Dataset**  | **Eval** | **NOSMOG**     | **RbM**        | **RbM(grad)**  |
> > |--------------|----------|----------------|----------------|----------------|
> > | OGB-ArXive   | ind      | 67.97$\pm$0.46 | 71.31$\pm$0.20 | 70.91$\pm$0.74 |
> > | OGB-ArXive   | tran     | 72.79$\pm$0.09 | 72.48$\pm$0.13 | 71.83$\pm$0.14 |
> > | OGB-Products | ind      | 77.29$\pm$0.71 | 80.88$\pm$0.24 | 80.75$\pm$0.35 |
> > | OGB-Products | tran     | 77.19$\pm$0.41 | 81.04$\pm$0.37 | 80.71$\pm$0.43 |

---

> > > ### Author Response · Authors · 2024-08-31
> > > **Official response to Reviewer SUMT (3/3)**
> > >
> > > **Q5.** It is difficult to determine exactly why increasing the number of parameters of MLP does not consistently lead to better performance. For many architectures, there is no guarantee that a brute-force increase of the number of parameters or hidden state dimension will improve performance. In the case of the proposed architecture, we see that different experts are focusing on different subtasks (discerning between a subset of the classes). The performance improvement we obtain is obtained partially by an ensemble effect, and partially by allowing different experts to learn easier tasks. The expanded MLP does not have either of these advantages. It is of interest that in some cases where the smaller MLP performs very poorly, the increase in the number of parameters leads to a very substantial improvement (e.g., GLNN - Products). When the small MLP's performance is already close to (or slightly better than) that of the teacher, the increase in the number of parameters has no significant positive impact. This suggests that there is some degree of saturation in the learning (the smaller MLP already has enough parameters to achieve the best performance possible for the adopted learning framework).
> > >
> > > The additional parameters do appear to help to compensate for an absence of positional encodings. Based on data from Tables 1 and 2, one can observe that GLNN and KRD (neither of which rely on positional encodings) both generally benefit from parameter inflation, while NOSMOG and CoHOP do not. NOSMOG adds DeepWalk positional encodings and CoHOP utilises label propagation, which can be viewed as a form of positional encoding.
> > >
> > > **Q6.** **Necessity of distillation and significance of student model.**
> > > In short, distillation is a technique that targets subsequent deployment, with the goal of reducing inference latency for a random node label query.
> > > The distillation setting is targeting two situations: (transductive) node features have changed; (inductive) new node(s) have been added.
> > > In both scenarios, a new prediction is necessary for the given node. If we are in a setting where fetching node features introduces substantial latency, then the aggregation step of a GNN can be very expensive. Large-scale industrial systems can face millions of queries per-second, with very frequent updates of node features, so it can be very important to reduce the response latency.
> > >
> > > The distilled models do not require features from any other nodes apart from the queried node. This leads to a substantial reduction in latency during operation. The papers proposing distillation architectures we cite demonstrate at least a factor of 10 reduction in terms of inference execution time for the MLP model versus the teacher GNN. Our computational load is slightly larger than the MLP, but still much smaller than the GraphSAGE (we discuss how the computation scales in an appendix).
> > >
> > > To make the necessity of distillation and the utility of a lightweight student model clearer, we have expanded the first paragraph of introduction.
> > >
> > > **Inductive setting.** Yes, we are assuming scenarios where a much larger number of unobserved test nodes will be added at a later time. Consider a transaction graph, for example, used for fraud detection -- nodes (transactions) are incrementally added to the graph. We use the setting of 20\% of test nodes to be comparable with the experimental set-up of other baselines.
> > >
> > > **Different graphs.** No, we do not consider a situation where the same model is applied to a different graph.
> > > We consider a setting where new nodes are added to a graph.
> > >
> > > **Improved accuracy.** While our model demonstrates improved accuracy compared to the GraphSAGE teacher, this is not the focus of the work. Our goal is to come as close to the performance of the teacher as possible. In the case of GraphSAGE, outperformance is possible because the proposed architecture makes better use of structural information (via the included positional encodings). For more advanced teacher models (Table 2), we do not observe outperformance, but we can achieve performance close to that of the teacher, without the need to access the features of neighbouring nodes.

---

> > > > ### Author Response · Authors · 2024-08-31
> > > > **References for the official response to Reviewer SUMT**
> > > >
> > > > **References**
> > > > 1. Kaipeng Zhang, Zhenqiang Li, Zhifeng Li, Wei Liu, and Yoichi Sato. Neural Routing by Memory. *NeurIPS*, 2021a.
> > > > 1. Shichang Zhang, Yozen Liu, Yizhou Sun, and Neil Shah. Graph-less Neural Networks: Teaching Old MLPs New Tricks Via Distillation. *ICLR*, 2021b.
> > > > 1. Yijun Tian, Chuxu Zhang, Zhichun Guo, Xiangliang Zhang, and Nitesh Chawla. NOSMOG: Learning Noise-robust and Structure-aware MLPs on Graphs. *NeurIPS*, 2022.
> > > > 1. Noam Shazeer, Azalia Mirhoseini, Krzysztof Maziarz, Andy Davis, Quoc Le, Geoffrey Hinton, and Jeff Dean.
> > > > Outrageously Large Neural Networks: The Sparsely-Gated Mixture-of-Experts Layer. *ICLR*, 2016.
> > > > 1. Zhihao Jia, Sina Lin, Rex Ying, Jiaxuan You, Jure Leskovec, and Alex Aiken. Redundancy-free Computation
> > > > for Graph Neural Networks. *SIGKDD*, 2020.
> > > > 1. Chenguang Zheng, Hongzhi Chen, Yuxuan Cheng, Zhezheng Song, Yifan Wu, Changji Li, James Cheng,
> > > > Hao Yang, and Shuai Zhang. ByteGNN: Efficient Graph Neural Network Training at Large Scale. *VLDB*, 2022.
> > > > 1. Guohao Li, Matthias Müller, Bernard Ghanem, and Vladlen Koltun. Training Graph Neural Networks with
> > > > 1000 Layers. *ICML*, 2021.
> > > > 1. Lei Zhang, Xiaodong Yan, Jianshan He, Ruopeng Li, and Wei Chu. DRGCN: Dynamic Evolving Initial
> > > > Residual for Deep Graph Convolutional Networks. *AAAI*, 2023.
> > > > 1. Lirong Wu, Haitao Lin, Yufei Huang, and Stan Z. Li. Quantifying the Knowledge in GNNs for Reliable Distillation into MLPs. *ICML*, 2023.

---

> > > > > ### Comment · Reviewer_SUMT · 2024-09-13
> > > > >
> > > > > Thank you for your detailed reply. I think the reply is convincing.
> > > > > Typo: After (12): '... in equation ()', in which the number is missing.

---

### Decision · Action_Editor_SB4D · 2024-09-21

**Recommendation:** Accept as is

**Comment:**

My decision is based on unanimous opinion of the reviewers, especially the accept rating on reviewer Reviewer SUMT, as i found this review of the highest quality of the three.

**Audience:**

Yes individuals in TMLR's audience expected to be interested in knowing the findings of this paper.

**Claims And Evidence:**

Yes,  claims made in the submission supported by evidence.